# ACCELERATING SPIKING NEURAL NETWORK TRAINING USING THE $d$-BLOCK MODEL

## ABSTRACT

There is a growing interest in using spiking neural networks (SNNs) to study the brain *in silico* and in emulating them on neuromorphic computers due to their lower energy consumption compared to artificial neural networks (ANNs). Significant progress has been made in directly training SNNs to perform on a par with ANNs in terms of accuracy. However, these methods are slow due to their sequential nature and require careful network regularisation to avoid overfitting. We propose a new SNN model, the $d$-block model, with stochastic absolute refractory periods and recurrent conductance latencies, which reduces the number of sequential computations using fast vectorised operations. Our model obtains accelerated training speeds and state-of-the-art performance across various neuromorphic datasets without the need for any regularisation and using fewer spikes compared to standard SNNs.

## 1 INTRODUCTION

Artificial neural Networks (ANNs) are ubiquitous in achieving state-of-the-art performance across various domains, such as image recognition (He et al., 2016), natural language processing (NLP) (Brown et al., 2020) and computer games (Silver et al., 2017; Vinyals et al., 2019). These networks have also proven useful for studying the brain due to their architectural similarities (Richards et al., 2019) and have further advanced our understanding of the computational processes underlying the visual and auditory system (Harper et al., 2016; Singer et al., 2018; Cadena et al., 2019; Francl & McDermott, 2022; Yamins & DiCarlo, 2016). However, ANNs have been criticised for their substantial energy demands resulting from their continued exponential growth in size (Strubell et al., 2019; Schwartz et al., 2020), as exemplified by the GPT language models scaling from 110 million to 1.5 billion to 175 billion parameters to deliver ever-improving advances across various NLP tasks (Radford et al., 2018; 2019; Brown et al., 2020). Furthermore, the applicability of ANNs to neuroscience is confined by their activation function, as the brain employs spikes rather than continuous-valued outputs used by ANN units.

Spiking neural networks (SNNs) are a type of binary neural network (Figure 1a), which overcome these challenges as they consume drastically less energy than ANNs when deployed on neuromorphic hardware (Wunderlich et al., 2019) and their biological realism makes them a favourable model for studying the brain *in silico* (Vogels et al., 2011; Denève & Machens, 2016). However, SNN training remains a challenging problem due to the non-differentiable binary activation function employed by the spiking neurons. This has historically resulted in solutions that impose constraints on the neurons, such as rate codes (O'Connor et al., 2013; Esser et al., 2015; Rueckauer et al., 2016), or only allowing neurons to spike at most once (Bohte et al., 2002; Mostafa, 2017; Comsa et al., 2020).

A recent proposal known as surrogate gradient training can overcome these limitations and has been shown to improve training on challenging datasets using models of increasing biological realism (Eshraghian et al., 2021). Surrogate gradient training replaces the undefined derivate of the neuron's activation function with a surrogate function and uses backpropagation through time (BPPT) for training, since SNNs are a particular form of recurrent neural network (RNN) (Neftci et al., 2019). SNNs thus experience many shortcomings associated with RNNs, such as their notably slow training times resulting from their sequential nature (Kuchaiev & Ginsburg, 2017; Vaswani et al., 2017). Furthermore, SNNs require multiple regularisation terms to avoid

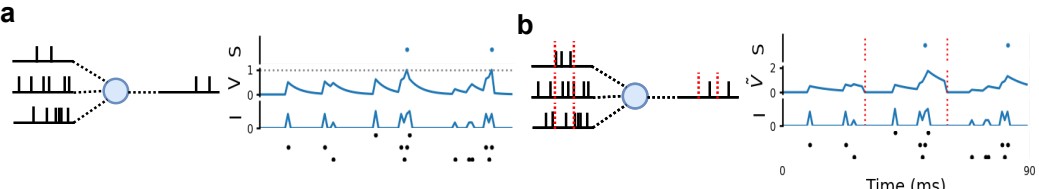

Figure 1: **a.** Left: The standard leaky integrate and fire (LIF) model. Right: Input and output activity of the neuron (bottom panel: Input raster, lower middle panel: Input current $I$, upper middle panel: Membrane potential $V$ with the dotted line representing the firing threshold, and top panel: Output spikes $S$). **b.** Left: Our $d$-block model (here $d = 3$ blocks with edges denoted by vertical dotted red lines). Right: Input and output activity of the neuron (same as in the LIF model but with a change in the upper middle panel, where the $d$-block computes membrane potentials without reset $\tilde{V}$).

overfitting and to obtain sparse spiking activity (Zenke & Vogels, 2021; Perez-Nieves & Goodman, 2021), where the latter is important in obtaining energy-efficient computations on neuromorphic hardware (Panda et al., 2020).

In this work, we address these shortcomings by proposing a new SNN model with stochastic absolute refractory periods and recurrent conductance latencies. The absolute refractory period is the brief period after a spike during which a neuron cannot spike again; it is a feature of the biology not typically incorporated into standard SNNs. In a recent study, Taylor et al. (2022) proposed a model - which we refer to as a block - for accelerated training of SNNs in which individual neurons spike at most once. We extend this model to being multi-spike and recurrent, by recurrently connecting and concatenating these blocks across time, and hence refer to our model as the $d$-block model, as each neuron can spike at most $d$ times when employing $d$ blocks (Figure 1b). Our contributions are summarised as follows.

1. We propose a new SNN model which uses fewer sequential operations compared to conventional SNNs and is henceforth more parallelisable on GPUs. We experimentally validate our model to obtain accelerated training speeds on synthetic benchmarks (up to 34× speedup) and various neuromorphic datasets (achieving up to 9× speedup).

2. Our model theoretically consumes less energy than conventional SNNs when employed on neuromorphic hardware, as it emits fewer spikes than standard SNNs during inference across multiple neuromorphic datasets. Furthermore, it does not include any training regularisation, thus avoiding the need for time-consuming and energy-demanding hyperparameter searches to find suitable regularisation values.

3. Notably we obtain state-of-the-art results on challenging neuromorphic datasets with an 86.20% and 68.16% accuracy on the Spiking Heidelberg Dataset (SHD) and Spiking Speech Commands (SSC) dataset respectively, raising performance by ~ 3% and ~ 8% over previously published results using standard SNNs.

## 2 BACKGROUND AND RELATED WORK

### 2.1 STANDARD SPIKING MODEL

A spiking neural network (SNN) is a type of recurrent neural network in which neurons output binary signals known as spikes. A neuron $i$ in layer $l$ (consisting of $\mathbb{R}^{N^{(l)}}$ neurons) emits a spike $S_i^{(l)}[t] = f(V_i^{(l)}[t]) \in \{0, 1\}$ at time $t$ if its membrane potential $V_i^{(l)}[t]$ reaches firing threshold $V_{th}$.

$$f(V_i^{(l)}[t]) = \begin{cases} 1, & \text{if } V_i^{(l)}[t] > V_{th} \\ 0, & \text{otherwise} \end{cases} \tag{1}$$

The leaky integrate and fire (LIF) model describes the evolution of the membrane potential $V_i^{(l)}(t)$ for resting potential $V_{rest} \in \mathbb{R}$, membrane time constant $\tau \in \mathbb{R}$ and input resistance $R \in \mathbb{R}$ (Gerstner et al., 2014).

$$\tau \frac{dV_i^{(l)}(t)}{dt} = -V_i^{(l)}(t) + V_{rest} + RI_i^{(l)}(t) \tag{2}$$

Without loss of generality the model is normalised ($V_i^{(l)}(t) \in [0,1]$ by $V_{rest} = 0, V_{th} = 1, R = 1$) and discretised (Taylor et al., 2022), such that for simulation time steps $t \in \{1,\dots,T\}$:

$$V_i^{(l)}[t+1] = \beta V_i^{(l)}[t] + (1-\beta)\underbrace{\left( b_i^{(l)} + \sum_{j=1}^{N^{(l-1)}} W_{ij}^{(l)} S_j^{(l-1)}[t+1] + \sum_{j=1}^{N^{(l)}} W_{ij}^{\text{rec }(l)} S_j^{(l)}[t] \right)}_{\text{Input current } I_i^{(l)}[t+1]} - \underbrace{S_i^{(l)}[t]}_{\text{Spike reset}} \tag{3}$$

The membrane potential is charged by the constant bias current source $b_i^{(l)}$, and the input $S^{(l-1)}[t] \in \mathbb{R}^{N^{(l-1)}}$ and output spikes $S^{(l)}[t-1] \in \mathbb{R}^{N^{(l)}}$ through feedforward $W^{(l)} \in \mathbb{R}^{N^{(l)} \times N^{(l-1)}}$ and recurrent connectivity $W^{\text{rec }(l)} \in \mathbb{R}^{N^{(l)} \times N^{(l)}}$ respectively. At every time step the membrane potential dissipates by a factor $0 \le \beta = \exp(\frac{-\Delta t}{\tau}) \le 1$ (for simulation time resolution $\Delta t \in \mathbb{R}$). The membrane potential is at rest $V_{rest} = 0$ in the absence of any input current and a spike $S_i^{(l)}[t] = 1$ is outputted if the potential rises above the firing threshold $V_{th} = 1$ (after which the potential is reduced back close to resting state).

## 2.2 TRAINING TECHNIQUES

To date, SNN training remains challenging. The success of backpropagation (Rumelhart et al., 1986) in ANNs does not naturally translate to SNNs due to their non-differentiable nature. Various methods have been proposed to circumvent this issue, however, these either fail to properly utilise the temporal dynamics of neurons or result in slow training times.

**Shadow training: ANN to SNN conversion**   Instead of training a SNN, shadow training converts an already trained ANN to a SNN such that the firing rates of the SNN neurons approximate the activations of the ANN units (O'Connor et al., 2013; Esser et al., 2015; Rueckauer et al., 2016; 2017). Although this method permits deep SNNs to perform well on challenging large-scale datasets like Imagenet (Deng et al., 2009), it endures various shortcomings. Firstly, it requires long simulation durations to obtain reasonable prediction accuracies, which have been argued to largely diminish the energy efficiencies of SNNs emulated on neuromorphic hardware (Davidson & Furber, 2021). Secondly, converted SNNs perform worse than ANNs due to the conversion process. Thirdly, although this performance gap can be reduced by coupling ANN and SNN training (Wu et al., 2021a;b; Kheradpisheh et al., 2021), the modelling applicability to neuroscience is limited due to the imposed rate-code (whereas the brain might instead process stimuli using a temporal code (Guyonneau et al., 2004; Cariani, 1999)).

**Direct SNN training**   Although the standard use of backpropagation is prohibited, alternative approaches have been proposed to estimate the gradients of network weights. Perturbation learning randomly perturbs network weights to approximate gradients, yet this method requires many trials to average out noise and scales poorly with growing network architectures (Williams, 1992; Seung, 2003). Latency learning calculates the gradients at the time of spiking as, unlike spikes, time is continuous (Bohte et al., 2002; Mostafa, 2017; Comsa et al., 2020; Kheradpisheh & Masquelier, 2020). However, this method enforces the modelling constraint that neurons spike at most once and is affected by the dead neuron problem, where a lack of spike activity is detrimental to learning. Surrogate gradient learning circumvents these issues by replacing the undefined derivative of the spike function with a surrogate function (Esser et al., 2016; Hunsberger & Eliasmith, 2015; Zenke & Ganguli, 2018; Lee et al., 2016). This method allows networks to learn temporal dynamics since it passes gradients through time (Bellec et al., 2018; Shrestha & Orchard, 2018; Neftci et al., 2019), and has thus become the status quo for directly training SNNs. However, learning is very slow as the network is sequentially simulated at every point in time (which we overcome by processing multiple time steps at once).

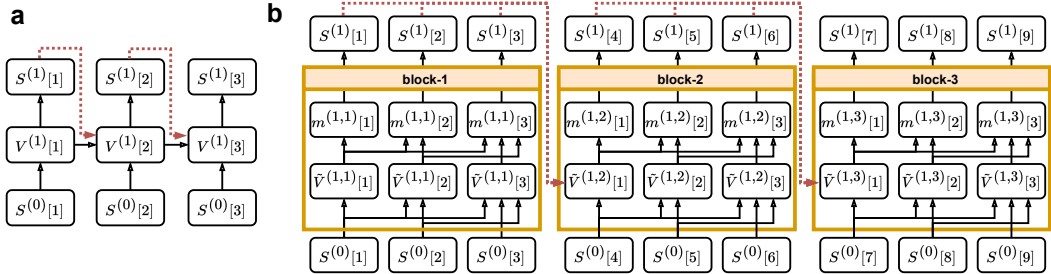

Figure 2: Computational graphs. **a.** Standard model. The membrane potentials $V^{(1)}[t]$ are recurrently dependent on the prior values $V^{(1)}[t-1]$ and charged by input spikes $S^{(0)}[t]$ and output spikes $S^{(1)}[t]$ through feedforward and recurrent weights (denoted by dotted red lines) respectively. Output spikes are generated by passing the membrane potentials $V^{(1)}[t]$ through the spike activation function. **b.** Our $d$-block model. Input spikes are processed by $d$ (here $d = 3$) equal length blocks, where every block is a single-spike SNN (where we have adopted the fast Taylor et al. (2022) model). Within layer $l$ and block $n$, input spikes $S^{(l-1)}[t]$ charge the membrane potentials without reset $\tilde{V}^{(l,n)}[t]$, which are mapped to spike outputs $S^{(l)}[t]$ using a fast grouping of operations $m^{(l,n)}[t]$ (see section 3.1). Recurrent connections (dotted red lines) transport output spikes from one block to another.

**Accelerated surrogate gradient training**   Some work has addressed the slow training times of SNNs using surrogate gradients. Perez-Nieves & Goodman (2021) managed to obtain faster training speeds by developing a sparse backprop algorithm, which only passes gradients through the membrane potentials that are close to the firing threshold. Although they manage to accelerate the backward pass up to 70×, these speedups depend on the development of custom CUDA kernels, which only support simple feedforward architectures (whereas we accelerate training of SNNs with recurrent connectivity and trainable neural time constants). Other works have managed to speed up training by removing the backward pass and performing all learning online (Bellec et al., 2020; Murray, 2019). These approaches, however, result in inferior performance accuracies on various datasets in comparison to standard surrogate gradient descent. Lastly, a core limitation of all of these speedup approaches is the prerequisite of sequentially simulating the network at every point in time (whereas we lessen this bottleneck).

## 3   FASTER TRAINING WITH THE $d$-BLOCK MODEL

We propose a new and fast algorithmic implementation for a SNN - called $d$-block - with stochastic absolute refractory periods and stochastic recurrent conductance latencies. We develop our model by extending the accelerated single-spike SNN by Taylor et al. (2022) from being single-spike and feedforward, to being multi-spike and recurrent.

### 3.1   THE SINGLE-SPIKE BLOCK MODEL

In the standard SNN, the input spikes $S^{(0)}[t]$ charge the membrane potentials $V^{(1)}[t]$ and output spikes $S^{(1)}[t]$ are emitted at every point in time $t$ (Figure 2a). This sequential processing is however slow. In a recent study Taylor et al. (2022) proposed a faster SNN model - with the assumption of single-spike outputs and feedforward connectivity - which avoids this sequential computation by instead computing all output spikes $S^{(1)}[t]$ simultaneously across time. This model, which we refer to as a block (orange box in Figure 2b), quickly transforms input spikes to output spikes using three steps:

1. Time series of presynaptic spikes $\mathbf{S}_j^{(l-1)}$ [1] are mapped to a time series of input currents $\mathbf{I}_i^{(l)}$ using a tensor multiplication $I_i^{(l)}[t] = \sum_{j=1}^{N^{(l-1)}} W_{ij}^{(l)} S_j^{(l-1)}[t]$.

---

[1]Bold face variables denotes arrays as opposed to scalar values.

2. Instead of computing membrane potentials $V^{(1)}[t]$ (as done in the standard model), the block calculates membrane potentials without reset $\tilde{V}^{(1)}[t]$ (by excluding the reset term in Equation 3), which are calculated using a fast convolution $\tilde{V}_i^{(l)}[t] = \beta^t V_i^{(l)}[0] + (1 - \beta)(\mathbf{I}_i^{(l)} \circledast \boldsymbol{\beta})[t]$ (where $\boldsymbol{\beta} = [\beta^0, \beta^1, \cdots, \beta^{T-1}]$).

3. Correct output spikes $\mathbf{S}_i^{(l)}$ are obtained using the following operations (which we group as $m$ in Figure 2). (1) Erroneous output spikes $\tilde{\mathbf{S}}_i^{(l)}$ are obtained by passing the no-reset membrane potentials $\tilde{\mathbf{V}}_i^{(l)}$ through the spike function $f$. (2) These erroneous output spikes are transformed into a latent encoding $\mathbf{z}_i^{(l)} = \sum_{k=1}^{t} \tilde{S}_i^{(l)}[k](t-k+1)$, where every element therein encodes an ordering of the spikes. (3) Correct output spikes are obtained through transformation $\mathbf{S}_i^{(l)} = g(\mathbf{z}_i^{(l)})$, where

$$g(\mathbf{z}_i^{(l)})[t] = \begin{cases} 1, & \text{if } z_i^{(l)}[t] = 1 \\ 0, & \text{otherwise} \end{cases} \tag{4}$$

## 3.2 EXTENDING THE SINGLE-SPIKE BLOCK TO MULTI-SPIKE BLOCKS

The block model only emits a single spike over the simulation period $T$. To extend the model to use multiple spikes, we divide the simulation period $T$ into $d$ chunks of equal length $T/d$ and process each chunk by a block using the same neural parameters (i.e. weight sharing the synaptic connectivity and membrane time constants; Figure 2b). If the input data is not divisible by $d$, we split the data into $d$ chunks of length $\lfloor T/d \rfloor$ and a final chunk of length $T - d \cdot \lfloor T/d \rfloor$. In this multi-block extension, which we refer to as the $d$-block model, neurons can spike up to $d$ times over the simulation period $T$. Every neuron is governed by a stochastic absolute refractory period as a neuron can spike at most once within a block of length $T/d$ and will therefore need to wait between 1 and $T/d$ time steps before it can spike again in the next block. From hereon forth we index variables of layer $l$ in block $n$ with superscript $(l, n)$ (e.g. the no-reset membrane potential value at time $t$ of neuron $i$ in layer $l$ and block $n$ is denoted as $\tilde{V}_i^{(l,n)}[t]$).

## 3.3 INCLUDING RECURRENT CONNECTIONS

The block model does not include any recurrent connectivity. Now that we have extended the model to be multi-spike, we can include recurrent connectivity by connecting the output spikes of block-$n$ as input to the next block-$(n+1)$ (see red lines in Figure 2). We achieve this by creating a binary vector of spike states (i.e. spike or no spike) for each neuron within a block and feeding these as input to the same neurons in the next block. Specifically, the output spikes $\mathbf{S}^{(l,n)} \in \mathbb{R}^{N^{(l)} \times T}$ in layer $l$ and block $n$ are flattened along time $\max_t S^{(l,n)}[t] \in \mathbb{R}^{N^{(l)}}$ (to denote if the neurons have spiked within the block's duration) before transporting them to the next block $n+1$ via the recurrent connectivity $W^{\text{rec }(l)}$.[2] Different from the standard SNN, these connections have a variable conduction latency equivalent in duration to the refractory periods of the afferent neurons in layer $l$ and block-$n$. The starting (no-reset) membrane potential of neuron $i$ in layer $l$ and block-$(n+1)$ is then computed from the flattened output spikes of block-$n$ and the first input spikes to block-$(n+1)$.

$$\tilde{V}_i^{(l,n+1)}[1] = \underbrace{\left( b_i^{(l)} + \sum_{j=1}^{N^{(l-1)}} W_{ij}^{(l)} S_j^{(l-1,n+1)}[1] \right)}_{\text{Feedforward current}} + \underbrace{\sum_{j=1}^{N^{(l)}} W_{ij}^{\text{rec }(l)} \max_t S^{(l,n)}[t]}_{\text{Recurrent current}} \tag{5}$$

## 3.4 TRAINING THE D-BLOCK MODEL

The d-block model is trained using backpropagation with surrogate gradients (Neftci et al., 2019), where we replace the undefined derivative of the spike function (Equation 1) with a fast sigmoid function $\frac{df_{sur}(V)}{dV} = (\beta_{\text{sur}}|V| + 1)^{-2}$ (Zenke & Ganguli, 2018), which has been shown to work well

---

[2]For simplicity we have omitted the batch dimension, which is included in our implementation and used in our experiments.

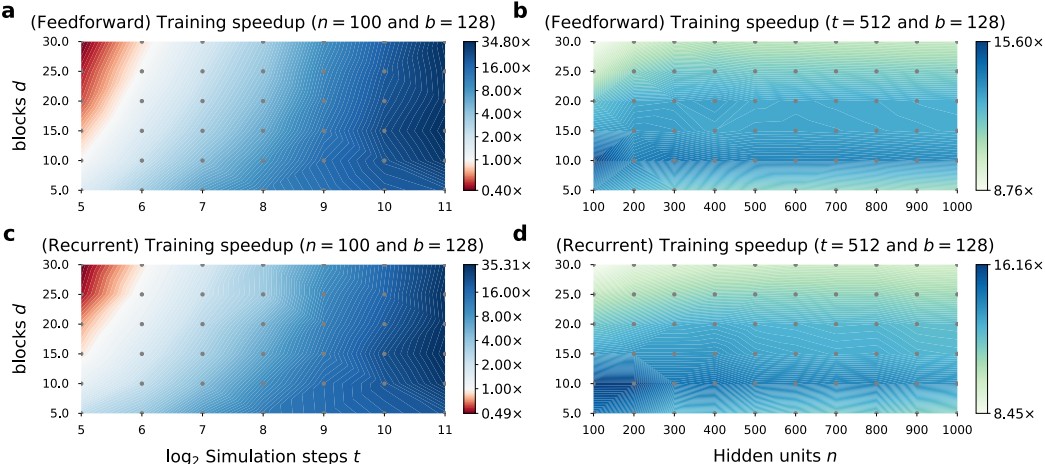

Figure 3: Training speedup of our $d$-block model over the standard SNN for feedforward and recurrent networks. **a.** Feedforward network training speedup as a function of the number of blocks $d$ and simulation steps $t$ (for fixed hidden neurons $n = 100$ and batch size $b = 128$). **b.** Feedforward network training speedup as a function of the number of blocks $d$ and hidden neurons $n$ (for fixed simulation steps $t = 512$ and batch size $b = 128$). **c.** Same as **a.** but for recurrent networks. **d.** Same as **b.** but for recurrent networks.

in practice (Zenke & Vogels, 2021) (here hyperparameter $\beta_{\text{sur}}$ defines the slope of the gradient). Exact training details for generating predictions, the loss function, optimisation procedures and hyperparameters can be found in the Appendix.

### 3.5 THEORETICAL SPEEDUP AND PERFORMANCE ADVANTAGES

Our model is more parallelisable than the standard SNN model. The computational complexity $O(NT)$ of a single neuron within the standard model is smaller than that of our model $O(dN\lfloor T/d \rfloor^2)$[3], for $N$ presynaptic neurons, $T$ simulation time steps and $d$ number of blocks. However, the sequential complexity of our model is $O(d)$ in comparison to the $O(T)$ complexity of the standard model. Thus, choosing $d$ smaller than $T$ and using GPUs - designed to execute many operations in parallel - leads to theoretical faster training speeds within our model.

We hypothesise the design of our model to be more robust to overfitting as - depending on the chosen number of blocks $d$ - our model emits less spikes, thus acting as a form of implicit regularisation on the spike code. In addition, the stochastic absolute refractory periods and stochastic recurrent conductance latencies act as a form of noise regularisation. Lastly, our model obtains a more salient flow of gradient through time due to the recurrent conductance latencies acting as a form skip connection (Srivastava et al., 2015; He et al., 2016), but through time.

## 4 EXPERIMENTS AND RESULTS

We evaluate the training speedup and performance of our model on real datasets in comparison to prior work. We used PyTorch (Paszke et al., 2017) for all implementations with benchmarks and training conducted on a cluster of NVIDIA Tesla V100 GPUs.

### 4.1 SPEEDUP BENCHMARKS

We benchmarked the time required to complete a single surrogate gradient training step in our $d$-block model and the standard SNN for a different number of blocks $d$, hidden units $n$, simulation steps $t$ and batch sizes $b$ on a synthetic spike dataset (see Appendix).

---

[3]The computational complexity $O(dN\lfloor T/d \rfloor^2)$ of our model comes from simulating $d$ blocks of length $\lfloor T/d \rfloor$, where a kernel of length $\lfloor T/d \rfloor$ is convolved over each of the $N$ presynaptic spike trains.

**Robust speedup for a different number of simulation steps**    Considering only feedforward SNNs (Figure 3a-b), we find our $d$-block model to train faster for a growing number of simulation steps, with a maximum training speedup of 34× for $d = 15$ blocks and $t = 2048$ simulation steps in a single hidden layer model of 100 feedforward neurons (Figure 3a). These speedups are robust over different numbers of blocks (e.g. we obtain speedups of 21×, 21× and 18× using $d = 10$, $d = 20$ and $d = 30$ blocks respectively over $t = 1024$ simulation steps), yet start to decline when the number of blocks approaches the number of simulation steps (e.g. we obtain a slower "speedup" of 0.4× when using $d = 30$ blocks and $t = 32$ simulation steps).[4] Speedups are further raised when using smaller batch sizes (with batch sizes $b = 32$ and $b = 64$ obtaining a maximum speedup of 40× and 37× respectively) or fixed membrane time constants (obtaining a maximum speedup of 44×; See Appendix).

**Robust speedups using different number of hidden units**    Also for feedforward SNNs, we find our $d$-block model to maintain robust training speedups over larger number of hidden units (e.g. obtaining a 13× speedup when using $n = 200$, $n = 400$ or $n = 600$ hidden units for $d = 10$ blocks and $t = 512$ simulation steps; Figure 3b). Again, these speedups are more pronounced when using smaller batch sizes (obtaining a 15× speedup when using $n = 200$, $n = 400$ or $n = 600$ hidden units for $d = 10$ blocks with batch size $b = 32$) or fixed membrane time constants (obtaining a 16× speedup when using $n = 200$, $n = 400$ or $n = 600$ hidden units for $d = 10$ blocks; see Appendix).

**Faster training speeds using recurrent connectivity**    Our recurrently connected model obtains faster training over the standard recurrent SNN. These speedups are slightly greater than the substantial speedups achieved when both models only employ feedforward connectivity. Again, in the recurrent case the training speeds increase with the number of simulation steps (with a maximum speedup of 35× for $d = 15$ blocks and $t = 2048$ simulation steps; Figure 3c) and are robust over larger numbers of hidden units (with an 18×, 17× and 17× speedup when using $n = 200$, $n = 400$ and $n = 600$ hidden units respectively for $d = 10$ blocks and $t = 512$ simulation steps; Figure 3d). As in the feedforward model, training speedups are further amplified in the recurrent model when using smaller batch sizes (with batch sizes $b = 32$ and $b = 64$ obtaining a maximum speedup of 41× and 39× respectively) or fixed membrane time constants (obtaining a maximum speedup of 44×; See Appendix).

## 4.2    PERFORMANCE ON REAL DATASETS

We tested the applicability of our model on different neuromorphic datasets of increasing difficulty. We confined ourselves to neuromorphic datasets - rather than image datasets - as these datasets are specifically crafted for SNNs, and best utilise their temporal and neural dynamics (*i.e.* leak and spike timing). The simplest dataset is the N-MNIST dataset (Orchard et al., 2015), in which the MNIST image dataset (LeCun, 1998) is converted into spikes using a neuromorphic vision sensor. More challenging are the Spiking Heidelberg Dataset (SHD) and Spiking Speech Commands (SSC) datasets, in which spoken digits and commands are converted into spikes using a model of auditory bushy cells in the cochlear nucleus (Cramer et al., 2020). These challenging temporal datasets provide a good base to evaluate our model as we are able to study the effect of using multiple blocks and recurrent connectivity, whereas other datasets like images are less suited as they are readily solved using single-spike and non-recurrent SNNs (Zhou et al., 2021).

**State-of-the-art results on neuromorphic datasets**    We obtain competitive accuracies across the different neuromorphic datasets, reaching an accuracy of 98.04%, 86.20% and 68.16% on the N-MNIST, SHD and SSC dataset respectively (Table 1). Notably, we improve performance on the challenging SHD and SSC dataset by ~ 3% and ~ 8% respectively over prior published results using standard recurrently connected LIF networks.

**Improving performance using recurrence, more blocks and additional layers**    We trained multiple models (comprising a single hidden layer of 128 neurons) across the different datasets to

---

[4]Our $d$-block model is identical to the standard SNN model when the number of blocks $d$ is equal to the number of simulation steps $t$, yet our model performs more computational work per time step and hence training slows down as $d$ tends to $t$ in the limit.

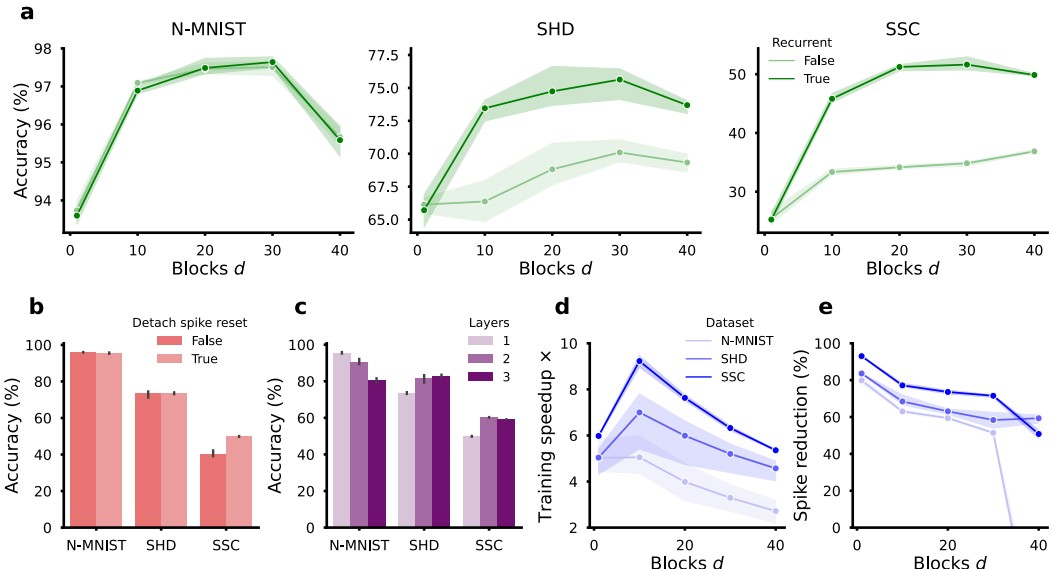

Figure 4: Analysis of our $d$-block model on challenging neuromorphic datasets. We use a single recurrently connected hidden layer network of 128 neurons and report results for three repeat runs of the model for which the mean and s.d. are plotted. **a.** Accuracy as a function of the number of blocks $d$ using feedforward and recurrent connectivity. **b.** Accuracy with the spike reset being attached or detached from the computational graph during training. **c.** Accuracy as a function of an increasing number of hidden layers. **d.** Training speedup of our model vs the standard SNN as a function of the number of blocks $d$. **e.** Reduction in spikes during inference of our model vs the standard SNN as a function of blocks $d$.

Table 1: Performance comparison to existing literature ($^\dagger$ denotes data augmentation, $^\beta$ denotes trainable time constants and we report our results in bold as an average of three repeat runs).

| Dataset | Model | Architecture | Neuron model | Accuracy (%) |
|---|---|---|---|---|
| N-MNIST | **our model** | 400-10 ($d = 30$) | recurrent LIF$^\beta$ | **98.04 ± 0.6** |
| | Shrestha & Orchard (2018) | 500-500-10 | feedforward LIF | 98.89 ± 0.06 |
| | Lee et al. (2016) | 800-10 | feedforward LIF | 98.66 |
| | Wu et al. (2018) | 400-400-10 | feedforward LIF | 98.78 |
| SHD | **our model** | 256-256-20 ($d = 30$) | recurrent LIF$^\beta$ | **86.20 ± 0.33** |
| | Cramer et al. (2020)$^\dagger$ | 1024-20 | recurrent LIF | 83.2 ± 1.30 |
| | Perez-Nieves et al. (2021) | 128-20 | recurrent LIF$^\beta$ | 82.7 ± 0.80 |
| | Eshraghian & Lu (2022) | 3000-20 | recurrent LIF | 82.27 ± 0.27 |
| | Zenke & Vogels (2021) | 256-20 | recurrent LIF | 82.0 ± 0.02 |
| | Cramer et al. (2022) | 186-20 | recurrent LIF | 76.2 ± 1.3 |
| | Cramer et al. (2020) | 128-20 | recurrent LIF | 71.4 ± 1.90 |
| SSC | **our model** | 256-256-256-35 ($d = 30$) | recurrent LIF$^\beta$ | **68.16 ± 0.28** |
| | Perez-Nieves et al. (2021) | 128-20 | recurrent LIF$^\beta$ | 60.1 ± 0.7 |
| | Cramer et al. (2020) | 128-20 | recurrent LIF | 50.9 ± 1.1 |

investigate the effect of including recurrence and increasing the number of blocks. We find that including recurrent connections in our model improves prediction accuracies across all datasets, especially on the more challenging SHD and SSC datasets (Figure 4a). Increasing the number of blocks also improves performance (Figure 4a). However, performance starts to drop for a large number of blocks ($d = 40$ across all datasets), likely due to overfitting as we do not include any regularisation. We also investigate the effect of detaching the flow of gradients through the recurrent connections, as doing so has been shown to improve training in standard recurrent SNNs (Zenke & Vogels, 2021). We observe no degradation in performance when detaching the spike reset terms from the computational graphs during training and find that it improves the performance on the SSC dataset (Figure 4b). Lastly, we investigate the effect of using additional hidden layers and find that additional layers improve accuracies on the SHD and SCC datasets, but degrade performance on the N-MNIST dataset (Figure 4c).

**Accelerated training** We find that our model trains faster than the standard SNN, with our model obtaining a maximum training speedup of ∼ 5×, ∼ 7×, and ∼ 9× on the N-MNIST, SHD, and SSC datasets, respectively, for $d = 10$ blocks (Figure 4d). The difference in speedups is due to the different temporal lengths and input and output dimensions of the datasets.

**Increased spike sparsity during inference** Our model uses fewer spikes than standard SNNs during inference, with a > 50% spike count reduction across all datasets when using $d = 30$ blocks and a > 80% reduction when using $d = 1$ block(Figure 4e). Our model thus theoretically requires less energy than standard SNNs if emulated on neuromorphic hardware, as the energy consumption scales approximately proportionally with the number of emitted spikes (Panda et al., 2020). However, we find that sparsity starts to decline for a growing number of blocks. Although we obtain favourable sparsity for the SHD (59% spike count reduction) and SSC (50% spike count reduction) datasets, we find our model to obtain worse sparsity on the N-MNIST dataset (41% spike count increase) when using $d = 40$ blocks. This suggests that sparsity advantages are dependent on the number of blocks and the dataset.

## 5 DISCUSSION

Surrogate gradient descent has been a substantial algorithmic development for directly training SNNs (Eshraghian et al., 2021). However, given the relation of SNNs to RNNs (Neftci et al., 2019), SNNs have been slow to train due to their sequential nature, precluding efficient parallelisation on GPUs (Kuchaiev & Ginsburg, 2017; Vaswani et al., 2017). In a recent study Taylor et al. (2022) proposed a model for accelerating the training of single-spike SNNs. However, due to the single-spike constraint, this model is less applicable to challenging neuromorphic datasets (compared to a recurrently connected multi-spike SNNs). In this work we address these shortcomings by extending their model to be multi-spike and recurrent, and experimentally validate this new model to achieve faster training speeds on both synthetic benchmarks (up to 34× speedup) and on various neuromorphic datasets (up to ∼ 9× speedup) compared to the standard multi-spike SNN. Furthermore, our model achieves state-of-the-art results on the challenging SHD and SSC neuromorphic datasets, with an accuracy of 86.20% and 68.16%, respectively, raising performance by ∼ 3% and ∼ 8% over prior published results using standard LIF networks. We are also able to do so without any training regularisation, whereas other works employ various regularisation constraints to avoid overfitting and to obtain an efficient spike code (Zenke & Vogels, 2021; Perez-Nieves & Goodman, 2021). Even so, we find our model to emit fewer spikes during inference in comparison to standard SNNs, thus theoretically lowering energy requirements relative to the standard SNN when emulating our model on neuromorphic hardware (Panda et al., 2020).

Our work can be improved in various avenues. Faster training speeds could be obtained by employing a sparse backprop implementation for our model, which has shown to be effective in standard SNNs (Perez-Nieves & Goodman, 2021). Here, backprop is modified to perform less computation by taking advantage of the sparse spiking nature of SNNs. The accuracy of our model could be improved by using different surrogate gradients (Zenke & Vogels, 2021), including additional modelling mechanisms such as temporal attention (Yao et al., 2021) or adaptive firing thresholds (Yin et al., 2021), or by using blocks of varying lengths (where we have limited ourselves to blocks of the same length).

## 6 REPRODUCIBILITY STATEMENT

We outline the theoretical construction of our model in section 3. An implementation of our model and instructions to replicate the experiments and results of this paper can be found at https://github.com/webstorms/DBlock, where we publish our code under the BSD 3-Clause Licence. Lastly, training details are provided in the Appendix.

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

## A APPENDIX

### A.1 SYNTHETIC SPIKE DATASET FOR THE SPEED BENCHMARKS

As outlined in (Taylor et al., 2022), we constructed binary input spikes of shape $B \times N \times T$ ($B$ being the batch size, $N$ the number of input neurons and $T$ the number of simulation steps), such that every batch dimension $b$ had a firing rate $r_b \sim \mathbf{U}(u_{\min}, u_{\max})$ uniformly sampled (with $u_{\min} = 0$Hz and $u_{\max} = 200$Hz). For every batch dimension we generated a random binary spike matrix of shape $N \times T$, such that every input neuron in the matrix had an expected firing rate of $r_b$Hz.

### A.2 TRAINING DETAILS AND HYPERPARAMETERS

#### A.2.1 READOUT NEURONS

Every network had an output layer of readout neurons (containing the same number of neurons as the number of classes within the dataset trained on), where we removed the spike and reset mechanism (as done in Zenke & Vogels (2021)). The output of readout neuron $c$ in response to input sample $c$ was taken to be the summated membrane potential over time $o_{b,c} = \sum_t V_{b,c}^L[t]$.

#### A.2.2 BETA CLIPPING

To enforce correct neuron dynamics we clipped the values of $\beta_i^{(l)}$ into the range $[0,1]$.

$$\beta_i^{(l)} = \begin{cases} 1, & \text{if } \beta_i^{(l)} > 1 \\ 0, & \text{if } \beta_i^{(l)} < 0 \end{cases} \tag{6}$$

#### A.2.3 WEIGHT INITIALISATION

All network connectivity weights were sampled from a uniform distribution $\mathbf{U}(-\sqrt{N^{-1}}, \sqrt{N^{-1}})$ where $N$ is the number of afferent connections. All biases were initialised as 0. All neurons in the hidden layers were initialised with a membrane time constant $\tau = 10$ms and $\tau = 20$ms for the readout neurons.

#### A.2.4 SUPERVISED TRAINING LOSS

We trained all networks to minimise a cross-entropy loss (with $B$ and $C$ being the number of batch samples and dataset classes respectively)

$$\mathscr{L} = -\frac{1}{B} \sum_{b=1}^{B} \sum_{c=1}^{C} y_{b,c} \log(p_{b,c}) \tag{7}$$

Variable $y_{b,c} \in \{0,1\}^C$ is the one hot target vector and $p_{b,c}$ are the network prediction probabilities, which were obtained by passing the readout neuron outputs $o_{b,c}$ through the softmax function.

$$p_{b,c} = \frac{\exp o_{b,c}}{\sum_{k=1}^{C} \exp o_{b,k}} \tag{8}$$

#### A.2.5 SURROGATE GRADIENT

We used the fast sigmoid function as our surrogate gradient (Zenke & Ganguli, 2018), which has been shown to work well in practice (Zenke & Vogels, 2021). Here hyperparameter $\beta_{\text{sur}}$ (which we set to 10 in all experiments) defines the slope of the gradient.

$$\frac{df_{sur}(V)}{dV} = (\beta_{\text{sur}}|V| + 1)^{-2} \tag{9}$$

#### A.2.6 TRAINING PROCEDURE

We used the Adam optimiser (with default parameters) (Kingma & Ba, 2014) for all training starting with an initial learning rate of 0.001, which was decayed by a factor of 10 every time the number of epochs reached a new milestone. A model was saved if it managed to lower the training error at the end of each epoch.

#### A.2.7 TRAINING HYPERPARAMETERS AND EXTENDED SPEEDUP RESULTS

Table 2: Dataset and corresponding training parameters for best performing models.

|  | N-MNIST | SHD | SSC |
| --- | --- | --- | --- |
| Dataset (train/test) | 60k/10k | 8156/2264 | 75466/20382 |
| Input neurons | 1156 | 700 | 700 |
| Dataset classes | 10 | 20 | 35 |
| Epochs | 50 | 60 | 80 |
| Learning rate | 0.001 | 0.001 | 0.001 |
| Batch size $B$ | 128 | 128 | 128 |
| Simulation steps $T$ | 300 | 500 | 500 |
| Time resolution $\Delta t$ (ms) | 1 | 2 | 2 |
| Milestones | (30) | (30) | (30, 60) |

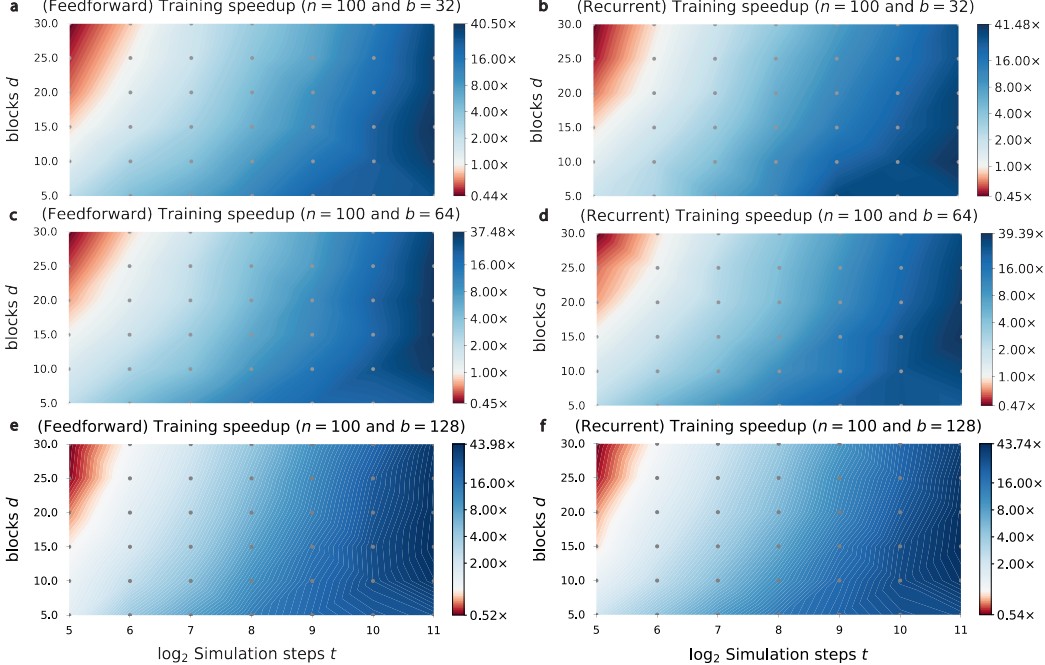

Figure 5: Training speedup of our $d$-block model over the standard SNN for feedforward and recurrent networks as a function of the number of blocks $d$ and simulation steps $t$ (for fixed hidden neurons $n = 100$). **a.-b.** Batch size 32. **c.-d.** Batch size 64. **e.-f.** Batch size 128 but using fixed membrane time constants. Left column: Feedforward networks and right column: Recurrent networks.

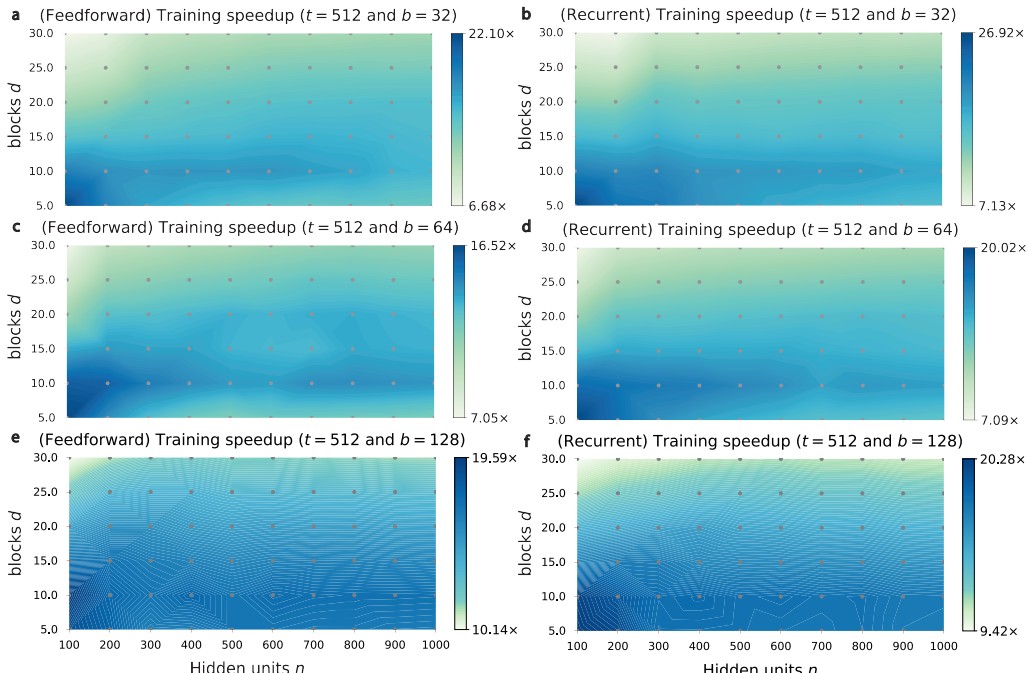

Figure 6: Training speedup of our $d$-block model over the standard SNN for feedforward and recurrent networks as a function of the number of blocks $d$ and hidden neurons $n$(for fixed simulation steps $t = 512$). **a.-b.** Batch size 32. **c.-d.** Batch size 64. **e.-f.** Batch size 128 but using fixed membrane time constants. Left column: Feedforward networks and right column: Recurrent networks.

