# OpenReview forum: "Accelerating spiking neural network training using the $d$-block model"
_ICLR.cc/2023/Conference — Submitted to ICLR 2023_

### Official Review · Reviewer_jrgn · 2022-10-21

**Confidence:** 3
**Correctness:** 3
**Technical Novelty And Significance:** 2
**Empirical Novelty And Significance:** 3
**Recommendation:** 3

**Clarity, Quality, Novelty And Reproducibility:**

Clarity:
In section 3.3, the author's interpretation of recurrent connections is unclear.
Novelty：
The author makes references about single-spike SNN proposed by Taylor et al. (2022). So the novelty in term of b-block itself is limited.


**Strength And Weaknesses:**

The authors didn’t give clear illustration and provide many detailed information to make readers to understand d-block model. The author makes references about single-spike SNN proposed by Taylor et al. (2022). But, they didn’t compare this work with single-spike SNN.

**Summary Of The Paper:**

This work propose a new SNN model named the d-block model, with stochastic absolute refractory periods and recurrent conductance latencies, which reduces the number of sequential computations using fast vectorised operations. Input spikes are processed by d equal length blocks, where every block is a single-spike SNN. This model obtains accelerated training speeds and state-of-the-art performance on the N-MNIST, SHD, and SSC datasets without the need for any regularization and using fewer spikes compared to standard SNNs. Besides, this model theoretically consumes less energy than conventional SNNs when employed on neuromorphic hardware.

**Summary Of The Review:**

I think this work is less innovative and the illustration of model isn’t clear.

---

### Official Review · Reviewer_Yn85 · 2022-10-24

**Confidence:** 4
**Correctness:** 2
**Technical Novelty And Significance:** 2
**Empirical Novelty And Significance:** 2
**Recommendation:** 3

**Clarity, Quality, Novelty And Reproducibility:**

Clarity: model presentation is not clear, and paper organization is poor.

Reproducibility: N/A

Significance/Quality/Novelty: Compared to Taylor et al.(2022), the novelty of the present work is very marginal given its marginal difference from the previous work. Thus, significance and quality of this work are quite poor.

**Strength And Weaknesses:**

Strength:

The idea to avoid the recursive update of membrane potential is of technical importance given the time and computational complexities of the recursive model which scale with the number of timesteps used. The authors successfully identified the feasibility of the proposed method on various event datasets.

Weaknesses:

1. The poor paper organization significantly undermines the readability. Also, the full explanation of the d-block model illustrated in 2b is not given. I do not still understand what $m$ and $\tilde{V}$ and their superscripts stand for.

2. Regarding the previous paper (Taylor et al. (2022)), I can find only very marginal difference of the present work from the previous work(Taylor et al. (2022)). The only difference is such that d (rather than one) spikes are used.

3. The previous works compared with the present work in Table 1 are out of dated. I recommend the authors to address SOTA results.

4. On Page 6, the authors address the github page for the code, which is against the double blind policy.


**Summary Of The Paper:**

The authors propose a method to control the number of output spikes (d-block model) and accelerate SNN calculations by avoiding the recursive calculation of membrane potential. The acceleration of SNN calculations was verified on various event datasets like N-MNIST, SHD, and SSC. As well, the model achieves high classification accuracy on the event datasets.

**Summary Of The Review:**

Overall, the significance of this work is limited given the marginal progress from the previous paper Taylor et al., (2022).

---

### Official Review · Reviewer_ddei · 2022-10-24

**Confidence:** 4
**Correctness:** 4
**Technical Novelty And Significance:** 2
**Empirical Novelty And Significance:** 3
**Recommendation:** 6

**Clarity, Quality, Novelty And Reproducibility:**

The paper is well-organized and clear. Related works were covered well. However, novelty is not strong enough. This paper is based on a modification of a previous work. The authors have made their source code public on GitHub, which is well-organized and reproducible.

**Strength And Weaknesses:**

Strength:
The proposed model achieves fewer sequential operations and lower energy consumption compared to conventional SNNs. They obtained SOTA accuracy scores on SHD (86.2%) and SSC (68.16%) are impressive.

Weakness:
1. The speedup experiments were conducted on small models. It is concerned that whether this method could work on larger and deeper SNNs, like ResNet.
2. There is limited novelty considering the reference (Taylor et al, 2022.) for accelerating training single-spike SNNs. It seems this article just makes incremental contributions.

**Summary Of The Paper:**

This paper proposes a new SNN model, named d-block model. By adding stochastic absolute refractory periods and recurrent conductance latencies, a d-block SNN can reduce the number of sequential computations using fast vectorized operations.

**Summary Of The Review:**

The paper proposes an efficient SNN training acceleration method. Novelty and experiments are limited.

---

### Official Review · Reviewer_D9Qe · 2022-10-27

**Confidence:** 4
**Correctness:** 3
**Technical Novelty And Significance:** 2
**Empirical Novelty And Significance:** 3
**Recommendation:** 5

**Clarity, Quality, Novelty And Reproducibility:**

I believe the novelty of this work is rather limited as it is based on a prior work. In terms of quality and clarity, the paper is decent. The code is also provided, which makes the paper reproducible while I haven't run the code myself.


**Strength And Weaknesses:**

Strengths:
-- The main strengths of the paper is its significant training time reduction.
-- The proposed model outperforms existing works in terms of performance.
-- The paper is well-written and east to understand.

Weaknesses:
-- The new model was built on top of an existing method by separating its architecture into multiple blocks. There is no mathematical foundation for the new model explaining why it should achieve a better results while its speedup during training is evident.
-- The architectures used for comparison are not the same which makes the comparison unfair and inconclusive. For example, it is expected to have better accuracy performance when a deeper model is used for SHD and SSC datasets in Table I.
-- Table I should also include training time per epoch for a fair comparison when using the same architecture.
-- I also expect to see some experimental results on more challenging datasets such as CIFAR10,100 and ImageNet.

**Summary Of The Paper:**

This paper presents a new architecture for spiking neural networks (SNNs), called d-block model. This architecture is built upon a previously proposed work (Taylor et al. 2022). The idea is to construct multiple blocks where within each there is dependency, allowing more parallelism during training. It has been shown that the new model outperform prior works both in terms of accuracy and also traiing time.


**Summary Of The Review:**

While the contribution of this paper is rather limited since it has been obtained by small modification to the prior work (Taylor et al. 2022), the training speedup is significant. The main issue that I have its comparison results in Table 1 where the networks architectures are not the same. I also expect to see the training time per epoch for a fair comparison.

---

### Official Review · Reviewer_b7YL · 2022-11-03

**Confidence:** 4
**Correctness:** 3
**Technical Novelty And Significance:** 2
**Empirical Novelty And Significance:** 3
**Recommendation:** 3

**Clarity, Quality, Novelty And Reproducibility:**

Clarity: The authors do not describe the used training methods. Audiences need to guess what they do.

Quality: The model achieves good performance and training efficiency. But it will be useless if it cannot be implemented on neuromorphic chips.

Novelty: This work is a naive extension of Taylor et al. (2022).

Reproducibility: Good.

**Strength And Weaknesses:**

Strength:

The proposed model indeed enables accelerated computing on GPU, and achieves sota results on some benchmarks.

Weaknesses:

1. This work is quite a naive extension of Taylor et al. (2022). If the authors of this work and Taylor et al. (2022) are the same, I highly recommend the authors to combine the two works into one.

2. The authors need to describe the training method explicitly in the main content. The proposed model can be treated as a modified LIF model, which is irrelevant to the training methods. From appendix A.3.5, which is not detailed, I guess that the existing surrogate gradient method is adopted. The accelerated training on GPU is due to the model's parallel computing nature, not due to some novel training method. The authors should make it clear.

3. Can the d-block model be implemented on neuromorphic hardware in an event-driven manner? First, the 1-block model is equivalent to the single-spike LIF model, so I do not worry about it. But for the d-block model, the spikes from the first 3 time steps are used in the 4th time step. Is it implementable? will it be implementable on future neuromorphic hardware?



**Summary Of The Paper:**

This work extends the 1-block model in Taylor et al. (2022) to the d-block model. Compared with the LIF model, the d-block model achieves accelerated computing on GPU by using fewer sequential operations.

**Summary Of The Review:**

If the authors want me to raise the score, they should convince me about the novelty and the implementability of the model.

---

### Decision · Program_Chairs · 2023-01-20

**Decision:**

Reject

**Justification For Why Not Higher Score:**

This paper should be rejected for sure.

**Justification For Why Not Lower Score:**

N/A

**Metareview: Summary, Strengths And Weaknesses:**

The paper got one 6 (marginally above threshold), one 5 (marginally below threshold) and three 3s (reject). The major challenges include lack of novelty and theoretical justifications, unsatisfactory paper writing, missing critical details, unconvincing experiments, etc. The authors did not submit a rebuttal. Moreover, reviewer Yn85 pointed out a possible violation of anonymity (github points to arXiv). So the AC recommended rejection.

**Summary Of Ac-Reviewer Meeting:**

N/A